# Visualized Hydraulic Fracture Re-Orientation in Directional Hydraulic Fracturing by Laboratory Experiments in Gelatin Samples

**Hua Zhang [1], Benben Liu [2] and Qingyuan He [2,\*]**

[1]  Department of Emergency Management of Shaanxi Province, Xi'an 710021, China
[2]  State Key Laboratory of Coal Resources and Safe Mining, School of Mines, China University of Mining and Technology, Xuzhou 221116, China; ts20020036a31@cumt.edu.cn
[\*]  Correspondence: 5783@cumt.edu.cn

**Abstract:** Directional hydraulic fracturing (DHF) is popular with hydraulic fracturing operations in coal mining to create cave-hard roofs, in which radial initial notches are created around open borehole walls before injecting high-pressurized fluid. Despite extensive field application of DHF, the three-dimensional irregular hydraulic fracture (HF) geometry in DHF remains unclear, and the HF re-orientation mechanism requires comprehensive understanding. Here, we experimentally examined factors affecting HF re-orientation in DHF in transparent gelatin samples with a self-developed experimental device. We found that it is the ratio between the differential stress and gelatin elastic moduls that determines HF re-orientation rather than the absolute magnitudes of these two factors. Both shear failure and tensile failure occur during HF re-orientation. The HF tends to propagate asymmetrically, and the step-like HF geometry is likely to form in gelatin samples with low elastic moduli and under high differential stresses. HF re-orientation is not necessarily a near-borehole effect, and HFs can propagate along the notch direction for longer distances in stiffer gelatin samples under relatively low or moderate differential stresses. Finally, recommendations are provided for the effective utilization of DHF at coal mine sites.

**Keywords:** directional hydraulic fracturing; hydraulic fracture re-orientation; hydraulic fracturing experiments in gelatin; three-dimensional irregular hydraulic fracture geometry

## 1. Introduction

Hydraulic fracturing has been widely applied in coal mining recently with various purposes, such as permeability enhancement of highly gassy coal seams and caveability control of hard roofs [1]. Theoretically, hydraulic fractures (HFs) are classified into two types based on their shapes and propagation planes relative to borehole axes, namely axial HFs (or termed longitudinal HFs, which are parallel to borehole axes) [2] and radial HFs (also called transverse HFs or penny-shaped HFs, which are perpendicular to borehole axes) [3]. The former HF type is common in the oil industry, and perforations must be generated around the cased borehole as artificial weaknesses (i.e., flow channels) for hydraulic fracturing and oil extraction [4]. Occasionally, the perforation direction is unparallel to the HF favorite propagation plane (that is, perpendicular to the minimum in situ stress direction) due to the local change of the in situ stress direction. Thus, axial HFs tend to re-orientate after their initiation, which usually leads to undesirable screen-out that blocks proppant transport [5]. On the contrary, open boreholes are popular with mining HFs. A specific cutting machine is used to produce a radial initial notch around the borehole wall as an artificial weakness for the initiation of a radial HF in the mining industry [6]. This technique is termed directional hydraulic fracturing (DHF) by many researchers [7–10]. The radial initial notch is expected to guide initiation and propagation of the HF in order to achieve orientation-controllable HFs in the rock mass (e.g., Figure 3 in Lekontsev and

Sazhin [7] and Figure 2 in Huang et al. [11]), since the HF orientation is the priority when HFs aim at inducing caving of hard roofs in coal mining [12]. Nevertheless, evidence from numerical modelling [13] and laboratory experiments [14] shows that the HF could undergo re-orientation after its directional initiation from the initial notch. Therefore, the HF re-orientation mechanism needs further comprehension for effective application of DHF at mine sites. This paper focuses its scope on identifying factors affecting DHF and revealing irregular three-dimensional HF geometries through laboratory experiments in totally transparent gelatin samples.

Re-orientation of axial HFs from oriented perforations has been well studied [15–17]. However, investigating the re-orientation of radial HFs becomes much more complicated because neither the plane strain assumption (which is commonly used to study the re-orientation of axial HFs [18,19]) nor the plane stress assumption are applicable to a radial-shaped HF. Hence, immature and time-consuming three-dimensional fluid-solid coupled numerical modelling must be performed to simulate DHF [13,20]. Most laboratory experiments on hydraulic fracturing are conducted in untransparent materials (e.g., rocks), and the existing experimental results of DHF provided by Deng et al. [14] only presented the final HF path from a two-dimensional section plane (see Figure 13 in Deng et al. [14]). The complete HF development process (i.e., HF initiation and propagation) and the tortuous three-dimensional HF geometry remain invisible. In addition, the simulation of the initial notch (which is the key part of DHF) in Deng et al. [14] is questionable. First, metal molds were used by Deng et al. [14] to simulate initial notches, which are much stiffer than their cement samples (having a Young's modulus of 8.4 GPa) and can hardly apply hydraulic pressure to mold–sample interfaces to induce the required stress concentration. Second, borehole axes in all the experiments in Deng et al. [14] were vertical, and notches (i.e., metal molds) were inclined to specific angles in different test scenarios. This differs from the fact that deviated boreholes are used at mine sites and initial notches are perpendicular to borehole axes.

In this study, we first use transparent gelatin samples to study DHF performed with deviated boreholes. A cutting tool is designed to create realistic radial weaknesses around borehole walls to simulate DHF in laboratory experiments. The complete HF initiation and propagation process is real-time recorded, and the three-dimensional HF geometry is observed directly and virtually through the transparent gelatin sample. Finally, factors affecting HF re-orientation from initial notches are qualitatively analyzed, and recommendations are given for effective application of DHF at mine sites.

## 2. Experimental Configuration

### 2.1. Related Works

Gelatin is homogeneous, isotropic, elastic, and brittle. It is recognized as an ideal analogue to rock-like materials in laboratory experiments (e.g., the Earth's upper crust) [21] and has been extensively used in geoscience, such as simulating magma intrusion and dyke propagation [22–24]. Hubbert and Willis [25] initiated hydraulic fracturing experiments in gelatin to demonstrate that HFs propagate perpendicularly to the minimum in situ stress direction. Then other researchers extended the application of gelatin in hydraulic fracturing studies with various objectives, such as examining the effect of buoyancy on HF growth [26], optimizing outside gravel pack techniques [27], observing the impacts of perforations and borehole orientations on HF initiation and propagation [28], and investigating factors affecting time-dependent HF geometries [29–31]. Ham and Kwon's [31] measurement results proved that the initiation pressure and fracture propagation velocity increased with the gelatin concentration. Moreover, the opening width of hydraulic fractures in gelatin is mainly determined by the medium stiffness and fluid pressure. In this study, gelatin is first used to simulate DHF, with specific focus on revealing factors influencing HF re-orientation and investigating three-dimensional irregular HF geometries through transparent gelatin samples.

### 2.2. Self-Developed Experimental Device

Figure 1 shows the schematic of the self-developed experimental device for studying DHF. The experimental device mainly consists of three parts: a container for placing gelatin samples, a pressurizing device, and a fracturing device. Each gelatin sample is cylindrical (with dimensions of 30 cm (in diameter) $\times$ 30 cm (in height)). The experimental container for the gelatin sample mainly consists of a polyvinyl chloride (PVC) pipe (with a diameter of 32 cm) and a cylinder-shaped polyolefin (POF) shrink film (with a diameter of 30 cm). Glass cement is squeezed into the gap between the PVC pipe and the POF shrink film (at the top and the bottom) to form a confined space. In each test, an air compressor is used to produce air pressure in this confined space in order to apply the confining stress to the gelatin sample [27,28]. The magnitude of the confining stress is monitored by a digital high-resolution (1 Pa) pressure gauge over the whole test. a horizontal radial confining stress is applied to the sample to induce the required differential stress for HF re-orientation (the confining stress in the vertical direction is zero) [25]. The gelatin type chosen for this experiment is 220 power, and the manufacturer is Fuyuan Gelatin Company. Weighing the specific weight concentrations of gelatin powder (i.e., 6%, 8%, and 10%) into about 80 °C hot water to fully melt into gelatin fluid. After gelatin fluid is poured into the experimental container (inside of the POF shrink film), a deviated plastic stick is put into the fluid and fixed by a wood plate. Then the experimental container is placed in a refrigerator for 24 h at 4 °C to solidify the gelatin fluid [31]. Afterwards, the plastic stick is removed from the solid gelatin sample to leave an open space in the sample as the hydraulic fracturing borehole (with a diameter of 10 mm). A steel cutting tool is used to create a radial initial notch around the borehole wall as an artificial weakness for DHF. The notch has a depth of 10 mm, which means the notch diameter is twice the borehole diameter [32]. Different from the metal molds used in previous studies (to simulate initial notches) [14], the steel cutting tool in this study creates a realistic radial notch in the gelatin sample, and the fluid pressure can directly act on the notch surfaces to effectively simulate the stress concentration induced by the initial notch in DHF. Then an injection tool is inserted into the open borehole, which includes a set of O-ring seals to simulate the hydraulic fracturing packer and several small holes as outlets for fracturing fluid [33]. The injection tool connects a syringe (with a maximum capacity of 500 mL) through a plastic tube. Plaster-of-paris slurry is taken as fracturing fluid since this material is allowed to set after hydraulic fracturing, thus providing a permanent record of HF geometries [25].

At this stage, the experimental device in this study basically has two limitations. First, the true triaxial confining stress condition is unrealized, and only a radial confining stress is applied to the gelatin sample in the horizontal direction. He et al. [12,34,35] demonstrated that HF re-orientation is dictated by differential stresses, and two different in situ stress conditions result in an identical HF re-orientation trajectory if they have the same differential stress magnitudes. Therefore, the gelatin sample in this study is radially confined in the horizontal direction to create the differential stress for DHF in a reverse faulting in situ stress state, which is common at block cave mines in Australia and coal mines in the northwest of China [36,37]. In addition, Hubbert and Willis [25] also hydraulically fractured a cylindrical gelatin sample sustaining radial confining pressure to examine the HF propagation plane relative to the minimum in situ stress direction. Second, the fracturing fluid in this study is manually injected into the gelatin sample rather than accurately controlled by a syringe pump [30,31]. It should be noted that high flow rate injection does enhance re-orientation of the HF. Hence, the manual injection of the plaster-of-paris slurry in each test is kept extremely slow to ensure HF propagation is under a quasi-static state, which is consistent with Takada's [26] hydraulic fracturing experiment.

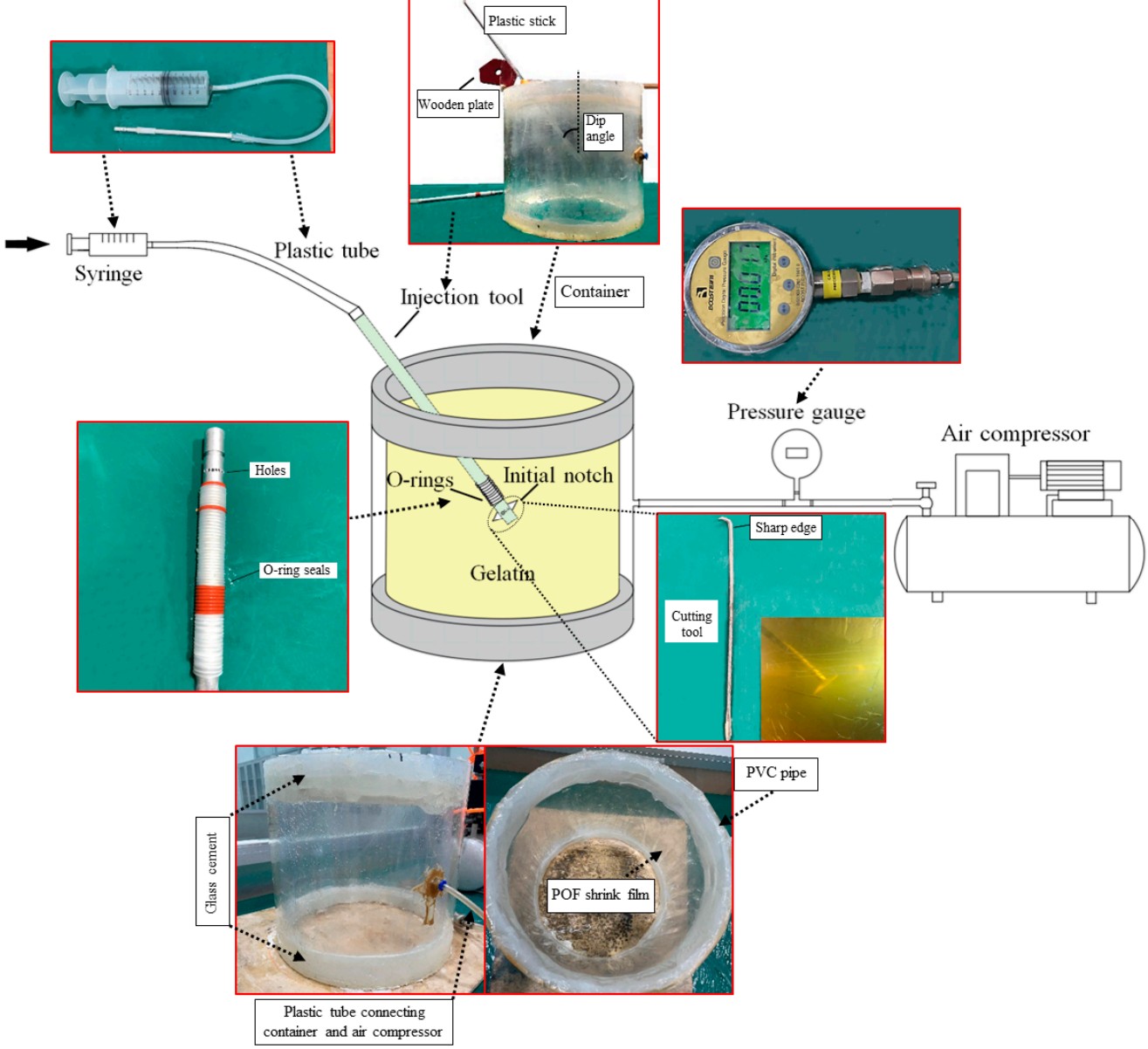

**Figure 1.** Schematic of experimental device.

## 3. Directional Hydraulic Fracturing Experiments in Gelatin Samples

### 3.1. Test Scenarios

Previous findings emphasized that differential stresses dominate HF re-orientation and HFs tend to propagate perpendicularly to the minimum in situ stress direction under high differential stresses [38]. Table 1 lists the differential stress magnitudes in some previous studies on HF re-orientation, which show their values ranging from 3.60 to 7.58 MPa. From the aspect of dimensional analysis [39], the effect of a dimensional factor (or factor group) on a physical system should be measured by another dimensional factor (or factor group) in this physical system that has the same unit [35]. Therefore, it might be the relative magnitude of the differential stress and another factor that determines re-orientation of the HF. In this section, this hypothesis is verified by a total of 9 laboratory experiments in gelatin samples, as listed in Table 2. In each test, the dip angle of the borehole is 45°, which is in a reasonable range for field-scale hydraulic fracturing at hard rock mines [40] and coal mines [41]. The experiments have three different test scenarios, in which the differential stress magnitudes are 1000 Pa, 1500 Pa, and 2000 Pa, respectively. These stress magnitudes are close to those in Stockhausen et al. [27] and Wu et al [28], in which hydraulic fracturing

is also conducted in gelatin samples under certain confining pressures (Table 3). In each test scenario, gelatin samples with three different weight concentrations (i.e., 6%, 8%, and 10%) are prepared [28,31] in order to investigate whether gelatin elastic moduli or gelatin strength influence DHF if the differential stress is kept constant.

**Table 1.** Differential stress magnitudes in previous hydraulic fracturing experiments in rock samples and cement.

| Mechanism of HF Re-Orientation | Magnitude of Differential Stress (MPa) | Reference |
|---|---|---|
| Oriented perforation | 7.58 | [38] |
| Stress shadow effect | 3.60 to 4.60 | [33] |
| DHF | 5.00 | [14] |

**Table 2.** Experimental schemes of DHF experiments in gelatin samples.

| Test Scenario | Test Case | Differential Stress (Pa) | Gelatin Concentration (%) | Elastic Modulus of Gelatin (Pa) |
|---|---|---|---|---|
| 1 | 1-1 | 1000 | 6 | 22,365 |
| | 1-2 | 1000 | 8 | 38,475 |
| | 1-3 | 1000 | 10 | 57,375 |
| 2 | 2-1 | 1500 | 6 | 22,365 |
| | 2-2 | 1500 | 8 | 38,475 |
| | 2-3 | 1500 | 10 | 57,375 |
| 3 | 3-1 | 2000 | 6 | 22,365 |
| | 3-2 | 2000 | 8 | 38,475 |
| | 3-3 | 2000 | 10 | 57,375 |

**Table 3.** Confining stress magnitudes in previous hydraulic fracturing experiments in gelatin samples.

| Magnitude of Confining Stress (Pa) | Reference |
|---|---|
| 1400 to 1700 | [27] |
| 1400 to 4800 | [28] |
| 1000 to 2000 | This study |

*3.2. Gelatin Elastic Moduli Measured by Indentation Tests*

As an indirect index of fracture propagation behavior, elastic modulus is a basic mechanical property required for analyzing fracture propagation behavior. Therefore, before DHF experiments, indentation tests are conducted to determine the elastic moduli of the gelatin samples (Figure 2). In the indentation test, each cylindrical gelatin specimen has dimensions of 8 cm (in diameter) × 12 cm (in height). A cylindrical steel indenter (with a diameter of 1 cm) is connected to a material testing machine to compress the gelatin specimen at a velocity of 5 mm/min [42] for 30 s [17]. The indenter diameter is smaller than one tenth of the specimen height, which minimizes the boundary effect in the indentation test [21,43]. For each gelatin concentration, five specimens are prepared, and their elastic moduli are separately measured by indentation tests. The highest and lowest values are excluded, and the average of the remaining three test results is calculated as the elastic modulus of the gelatin sample at the given concentration [44]. The final results are listed in Table 2, which show that the elastic modulus of the gelatin increases almost linearly with its concentration. Table 4 gives Ham and Kwon's [31] measurement results of the elastic moduli and fracture toughness of gelatin at different concentrations. It indicates that the fracture toughness of gelatin also changes linearly with its elastic modulus (see the linear regression in Table 4). This means, in each test scenario in Table 2, HF re-orientation trajectories in gelatin samples with different elastic moduli and strengths can be compared. The effect of the gelatin elastic modulus on HF re-orientation will be studied in the following contents.

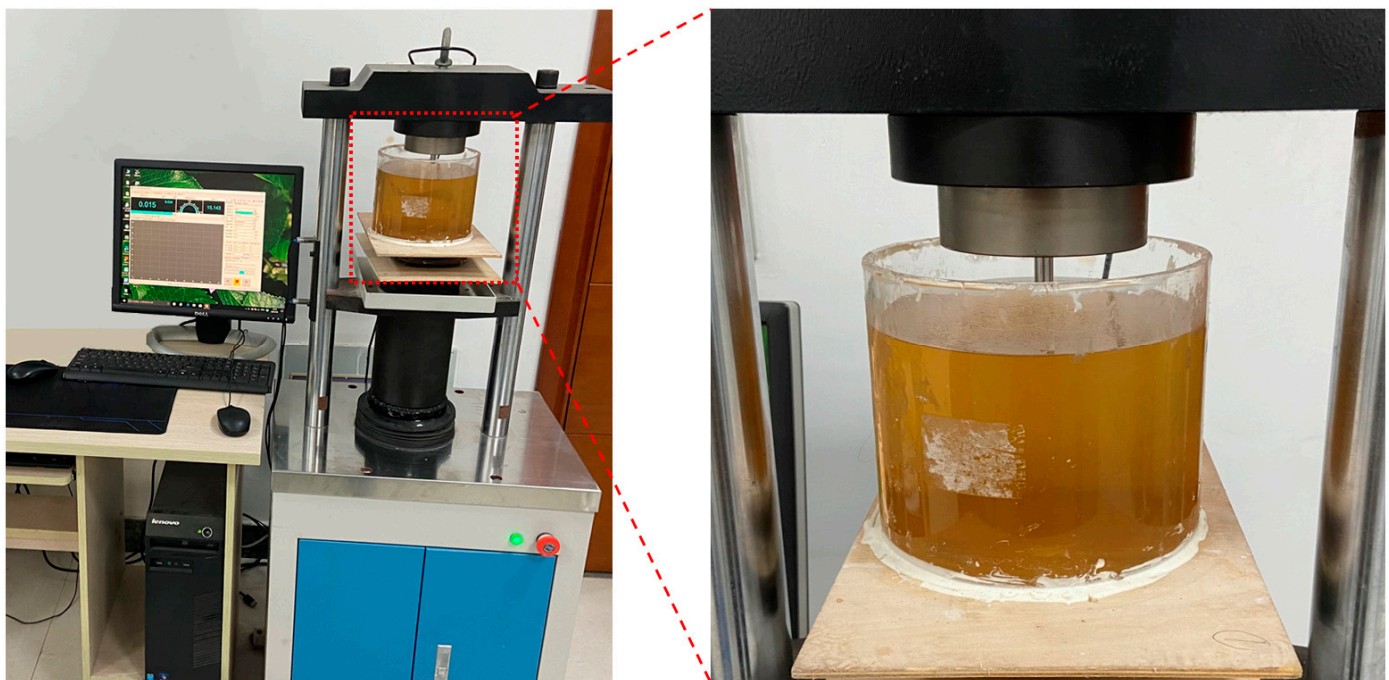

**Figure 2.** Indentation test of gelatin specimen.

**Table 4.** Elastic moduli and fracture toughness of gelatin at different concentration measured by Ham and Kwon [31].

| Gelatin Concentration (%) | Elastic Modulus of Gelatin/$E$ (Pa) | Fracture Toughness of Gelatin/$K_c$ (Pa · $\sqrt{m}$) |
|:---:|:---:|:---:|
| 7.4 | 38,000 | 400 |
| 12.3 | 90,000 | 1330 |
| 13.8 | 110,000 | 1530 |
| 16.7 | 283,000 | 3190 |
| Linear regression result | $K_c = 0.0108E + 205.68$ ($R^2 = 0.9768$) | |

*3.3. Results and Discussions*

Figures 3–5 provide the experimental results of Test Scenario 1 (Table 2), in which the gelatin concentration ($w$) increases from 6% to 10% and the differential stress ($\sigma_d$) is constant at 1000 Pa. The steel cutting tool effectively creates a radial initial notch around the open borehole wall as an artificial weakness for HF initiation (Figure 3a,b). Then the HF undergoes re-orientation and tends to propagate perpendicularly to the minimum in situ stress direction until it reaches the boundary of the experimental container (Figure 3c–e). Figure 3f shows the hydraulically fractured gelatin sample taken out from the container after the experiment. The sample surface looks uneven, and the guidance of the initial notch on HF propagation under this experimental condition ($w$ = 6% and $\sigma_d$ = 1000 Pa) is within a distance of about 5 to 6 times the borehole diameter. This indicates that HF re-orientation in DHF is not necessarily a near-borehole effect (i.e., within a distance twice the borehole diameter) [13], and the HF can propagate along the notch direction for a longer distance, depending on the experimental condition. Note that the differential stress in Test Case 1-1 is only 1000 Pa (which is much smaller than those in Table 1), and the HF still re-orientates after its directional initiation. Therefore, it could be the magnitude of the differential stress relative to the magnitude of another factor that determines the HF re-orientation distance.

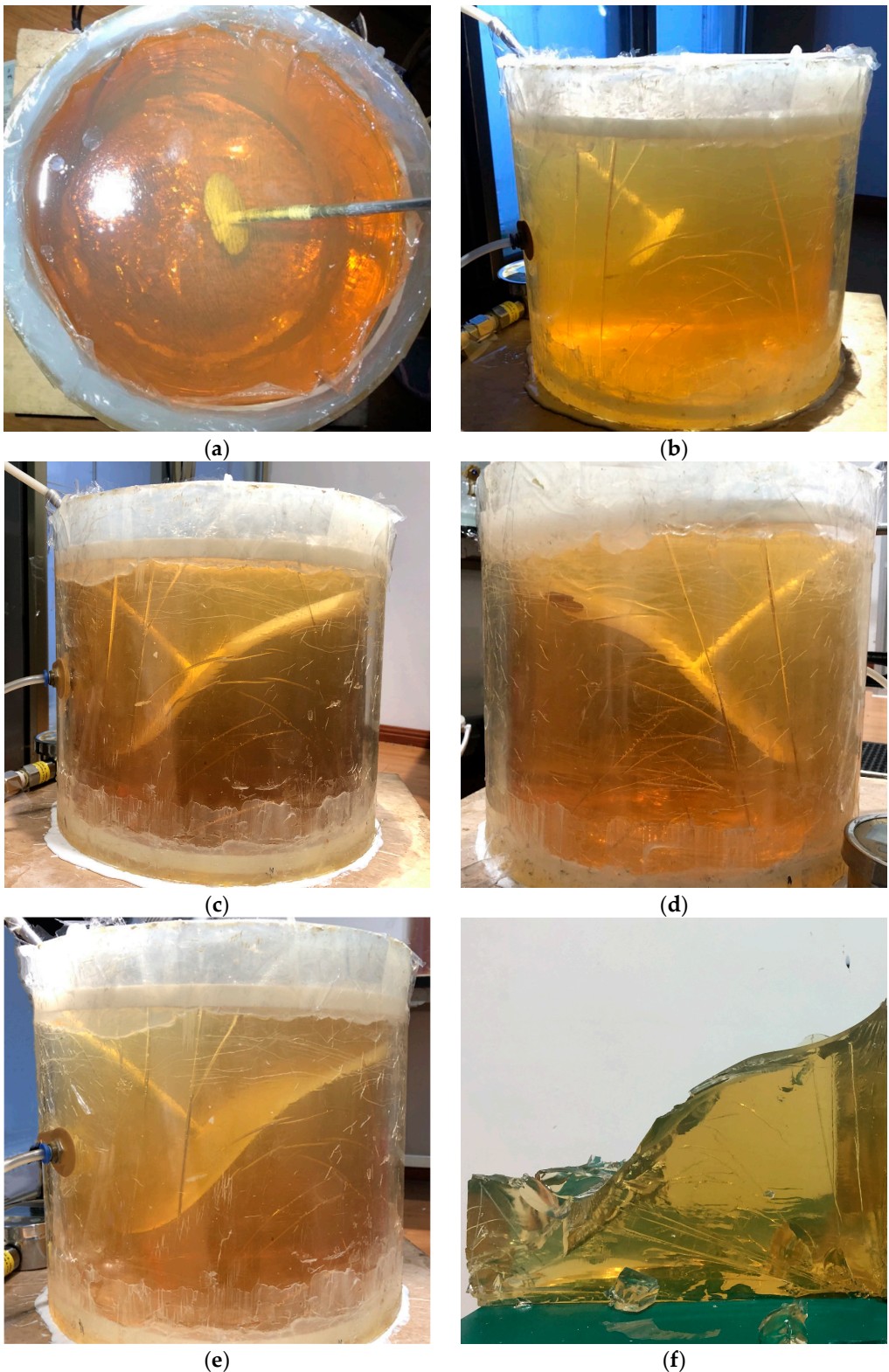

**Figure 3.** Results of Test Case 1-1 ($w$ = 6% and $\sigma_d$ = 1000 Pa): (**a**) HF initiates from initial notch (plan view); (**b**) HF initiates from initial notch (section view); (**c**) HF propagates along notch direction for certain distance before re-orientation; (**d**) HF re-orientation observed from another angle; (**e**) irregular HF geometry; (**f**) fractured gelatin sample taken out from container.

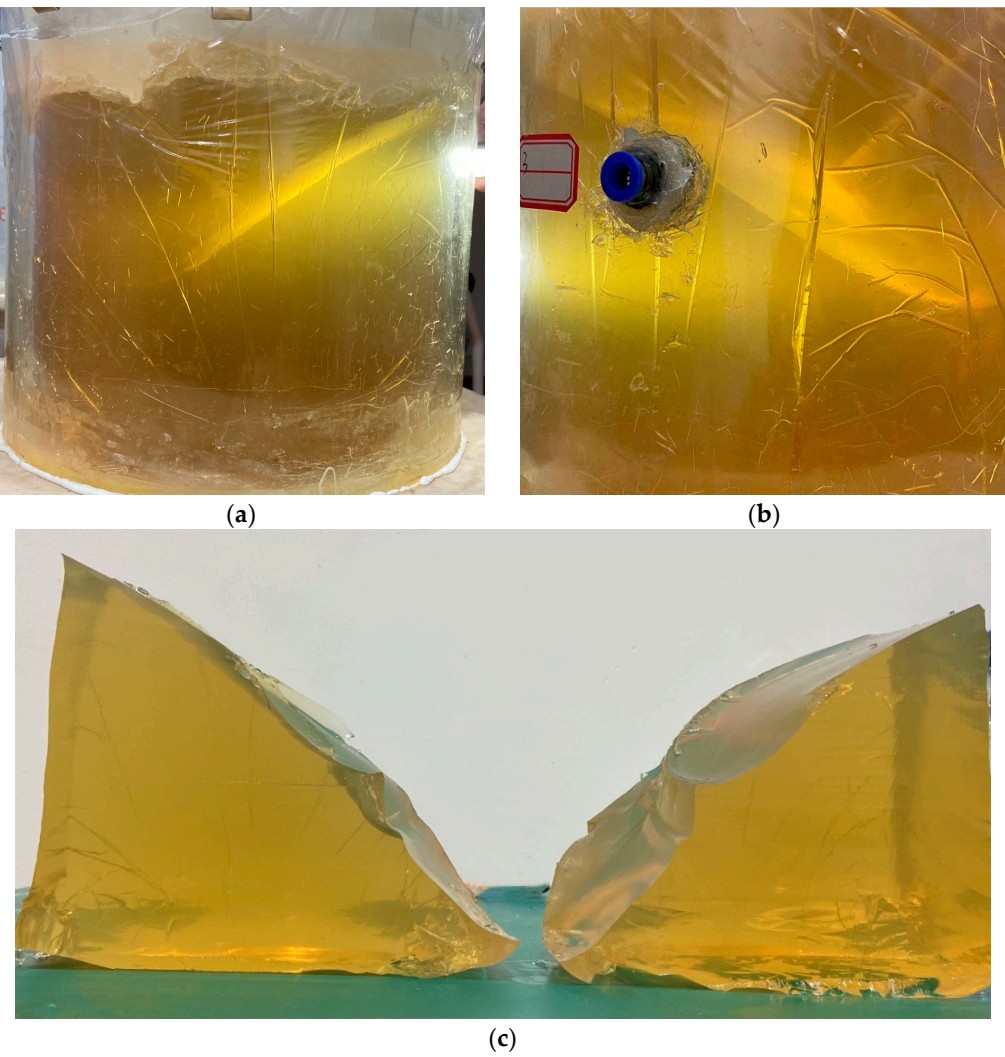

**Figure 4.** Results of Test Case 1-2 ($w$ = 8% and $\sigma_\mathrm{d}$ = 1000 Pa): (**a**) HF propagates along notch direction for certain distance before re-orientation; (**b**) HF re-orientation observed from another angle; (**c**) fractured gelatin sample taken out from container.

The gelatin concentration in Test Case 1-2 increases from 6% to 8% to examine the influence of the gelatin elastic modulus on DHF. Figure 4a,b show the HF trajectory that maintains a longer propagation distance along the notch direction until it re-orientates horizontally near the container boundary. The fracture surface of the sample in Test Case 1-2 (Figure 4c) becomes smoother compared with the S-shaped curved sample surface in Test Case 1-1 (Figure 3f). This demonstrates that the sample elastic modulus is another important factor affecting DHF, and the HF re-orientates more slowly towards its favorite propagation plane in a stiffer gelatin sample. The gelatin concentration further increases to 10% in Test Case 1-3, and the experimental results are provided in Figure 5. Figure 5a,b show the HF trajectories recorded from different angles during the DHF experiment, while Figure 5c,d give closer observation of the fractured sample that is taken out from the container after the experiment. The experimental device in this study is effective in presenting the complete HF initiation and propagation process through clear gelatin samples and the transparent container (Figure 5a,b), and the use of the thin POF shrink film allows convenient extraction of the fractured sample for detailed recoding of the final HF trajectory after the experiment (Figure 5c,d). The fractured sample surfaces in Test Case 1-3 ($w$ = 10%) are oblique (Figure 5e), which means HF propagation is more likely to be directionally controlled by the initial notch in a stiffer gelatin sample under a given differential stress condition ($\sigma_\mathrm{d}$ = 1000 Pa).

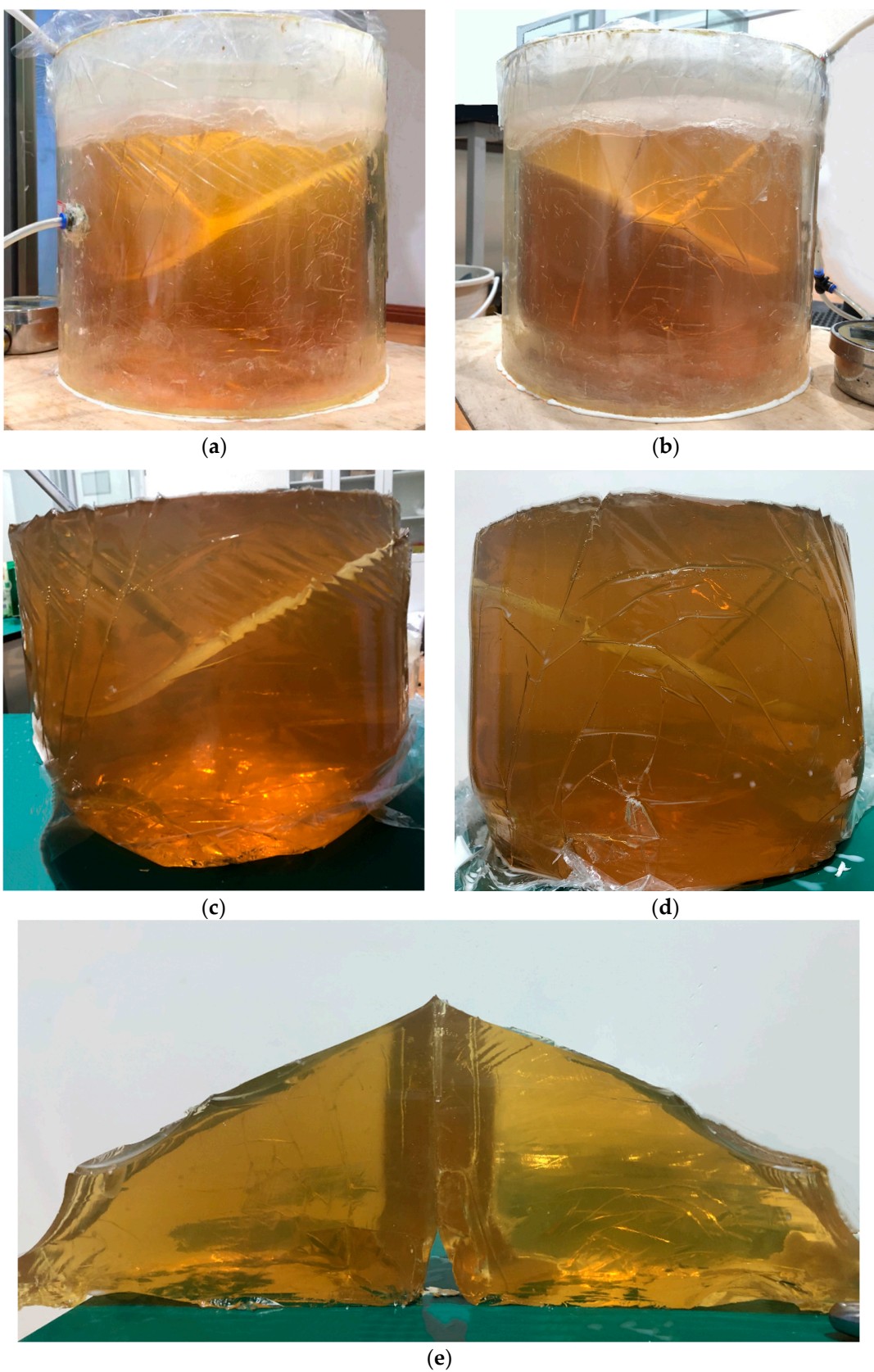

**Figure 5.** Results of Test Case 1-3 ($w$ = 10% and $\sigma_d$ = 1000 Pa): (**a**) HF propagates along notch direction; (**b**) HF trajectory observed from another angle; (**c**) fractured gelatin sample observed from the angle in Figure 5a; (**d**) fractured gelatin sample observed from the angle in Figure 5b; (**e**) fracture surfaces of gelatin sample.

In Test Scenario 2, the differential stress magnitude changes from 1000 Pa to 1500 Pa to investigate HF re-orientation in gelatin samples with different concentrations under a higher differential stress condition. Figure 6 shows the HF trajectory in Test Case 2-1 ($w$ = 6% and $\sigma_d$ = 1500 Pa). Compared with that in Test Case 1-1 ($w$ = 6% and $\sigma_d$ = 1000 Pa) (Figure 3c–e), the HF propagation plane looks asymmetric and re-orientates more quickly towards the horizontal direction (Figure 6a–c). Note that the difference between the differential stress magnitudes in Test Case 1-1 and Test Case 2-1 is only 500 Pa. This corroborates the hypothesis that the impact of the differential stress on DHF depends on its relative magnitude (e.g., relative to the gelatin elastic modulus) rather than its absolute magnitude. The existence of the differential stress restricts directional propagation of the HF along the initial notch, and the re-orientation distance is determined by the ratio between the differential stress and the gelatin elastic modulus. In addition, the three-dimensional HF geometry is not definitely as symmetric as that in numerical modelling with the homogeneous assumption [18]. The HF geometry tends to become asymmetric in a relatively high differential stress state. Note that gelatin is also a homogenous material, and hence the asymmetric propagation is not due to its heterogeneity. In Test Case 2-2, the gelatin concentration increases to 8%, and the differential stress is held at 1500 Pa (Figure 7). The HF geometry returns to a more symmetric pattern, and the initial notch again gives certain guidance on the HF propagation direction. This phenomenon becomes more obvious in Test Case 2-3, in which the gelation concentration is 10% (Figure 8). Also, it is found that the sample surfaces in these two test cases resemble each other (Figures 7e and 8e). This indicates that a higher gelatin elastic modulus ($w$ = 8% and $w$ = 10%) does prevent asymmetric propagation of the HF, but its promotion of DHF can be limited if the differential stress is sufficiently high (compared with that in Test Scenario 1).

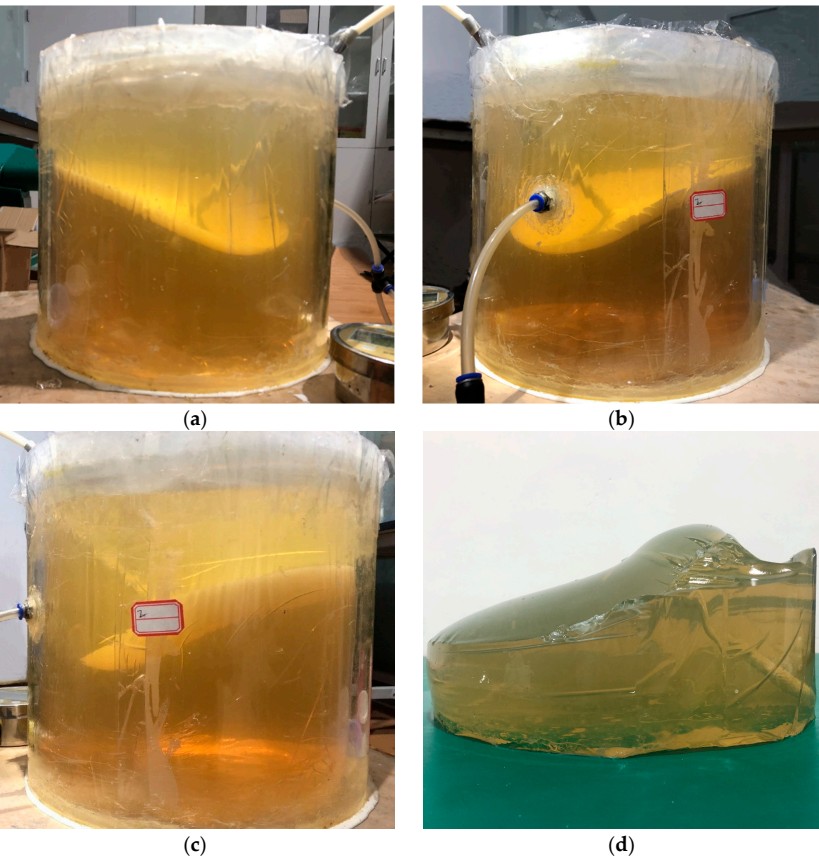

**Figure 6.** Results of Test Case 2-1 ($w$ = 6% and $\sigma_d$ = 1500 Pa): (**a**) HF re-orientates towards its favorite propagation plane; (**b**) HF re-orientation observed from another angle; (**c**) asymmetrical HF propagation; (**d**) fractured gelatin sample taken out from container.

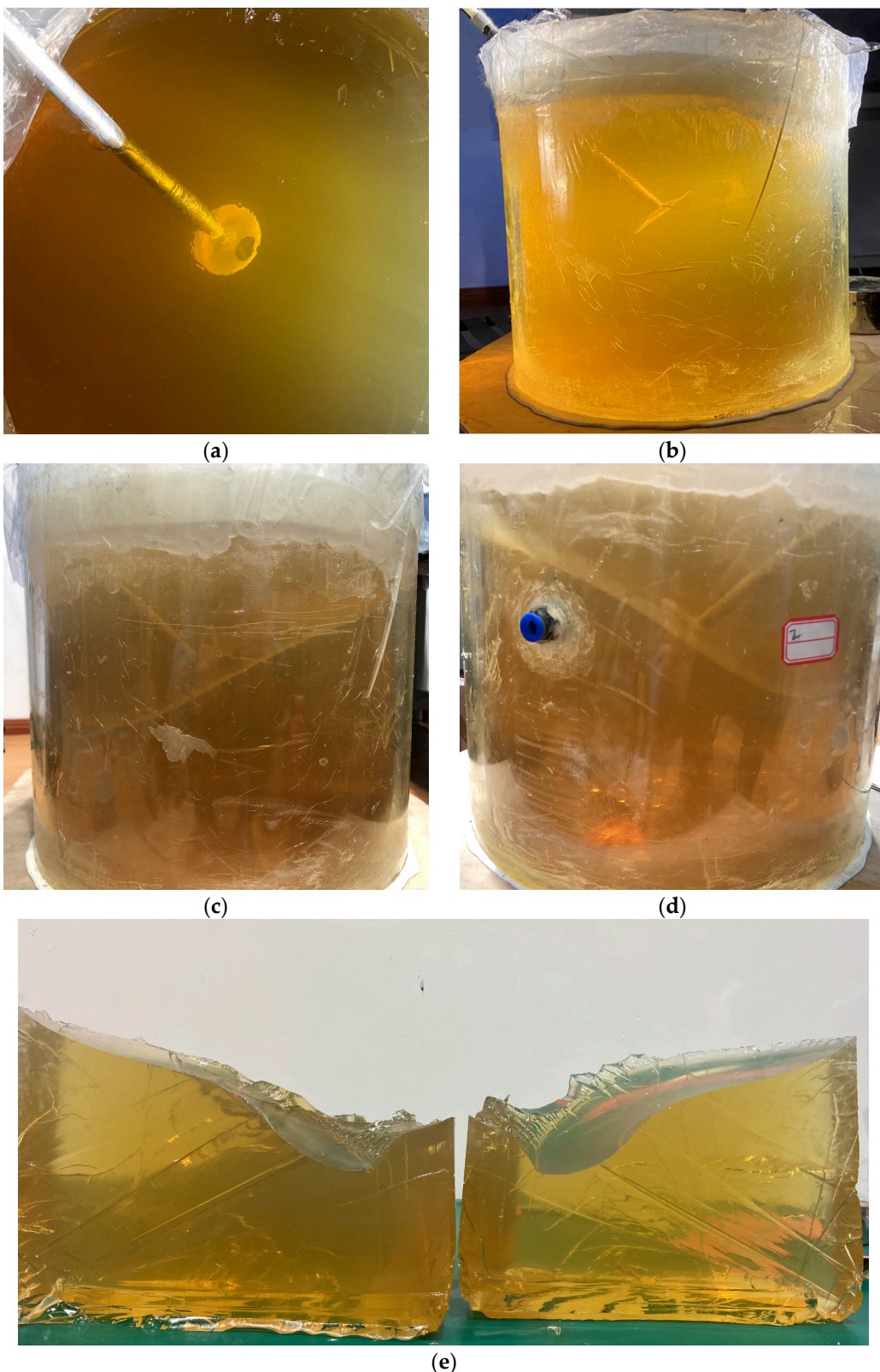

**Figure 7.** Results of Test Case 2-2 ($w$ = 8% and $\sigma_{\mathrm{d}}$ = 1500 Pa): (**a**) HF initiates from initial notch (plan view); (**b**) HF initiates from initial notch (section view); (**c**) HF propagates obliquely with limited re-orientation; (**d**) HF trajectory observed from another angle; (**e**) fractured gelatin sample taken out from container.

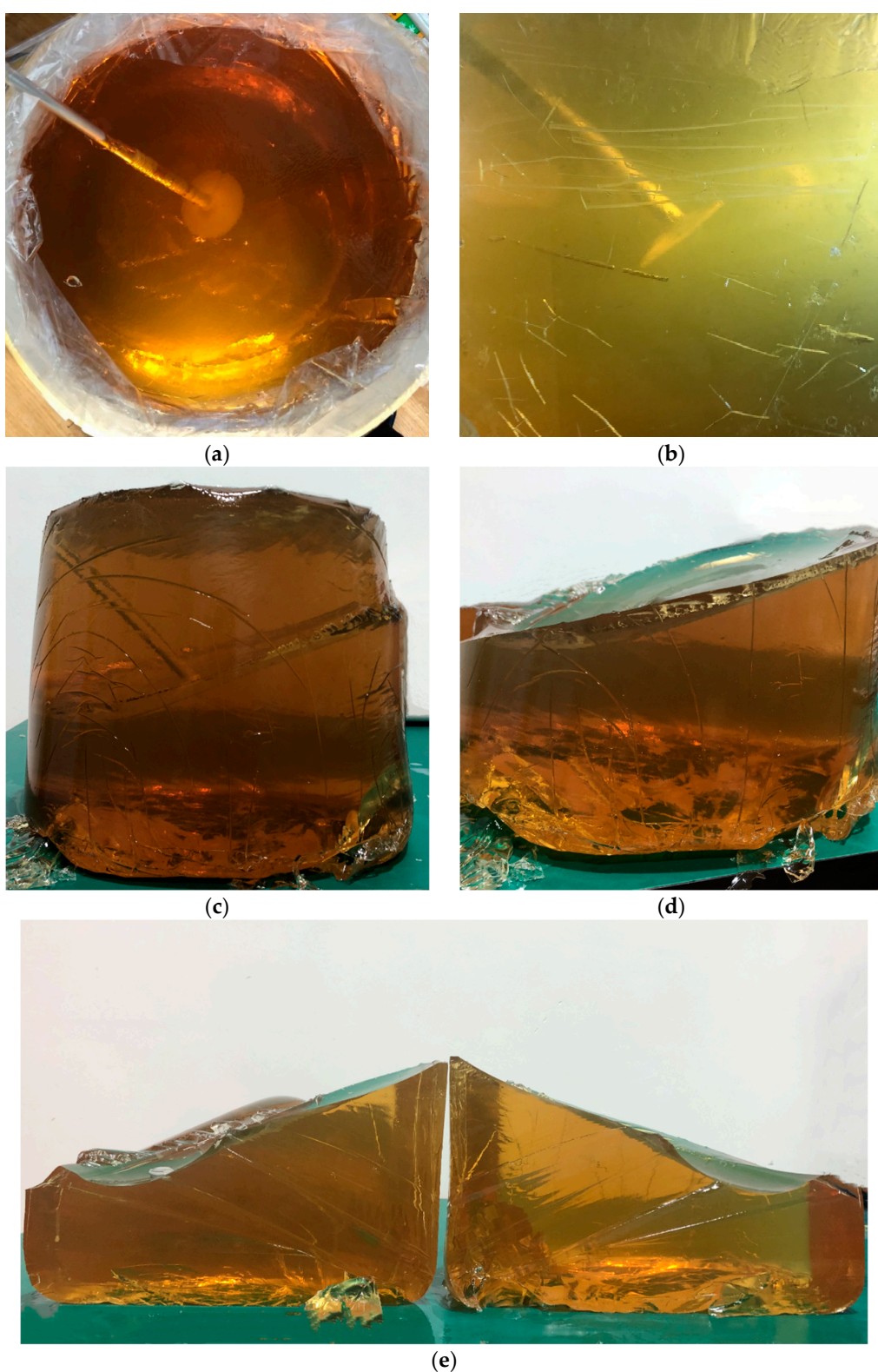

**Figure 8.** Results of Test Case 2-3 ($w$ = 10% and $\sigma_d$ = 1500 Pa): (**a**) HF initiates from initial notch (plan view); (**b**) HF initiates from initial notch (section view); (**c**) HF propagates obliquely; (**d**) smooth fracture surface; (**e**) fractured gelatin sample taken out from container.

The differential stress in Test Scenario 3 grows to 2000 Pa to provide further support for the above conclusions. The HF geometry in Test Case 3-1 ($w$ = 6% and $\sigma$d = 2000 Pa) shows a very irregular, asymmetric pattern (Figure 9b–d) even if the HF directionally

initiates from the initial notch (Figure 9a,e). A step-like HF geometry is noticed in Figure 9d, which is consistent with Daneshy and Services's [45] and Abass et al.'s [46] findings of HF re-orientation from oriented perforations. This implies that both shear failure and tensile failure occur during HF re-orientation, and hence shear strength of a material must be considered in the numerical simulation of DHF [13]. Otherwise, the HF re-orientation distance could be overestimated, as in Sepehri et al.'s [18] study, which only includes the tensile fracture toughness of the material. Nevertheless, it should be noted that formation of step-like HFs is not a necessity in DHF (e.g., Figure 6). The HF shape depends on the differential stress state and also the material elastic modulus (or material strength). In all the test cases, steps are only found in Test Case 3-1, which has the lowest gelatin concentration ($w$ = 6%) and the highest differential stress ($\sigma_d$ = 2000 Pa). In the other two test cases ($w$ = 8% and $w$ = 10%) in Test Scenario 3 ($\sigma_d$ = 2000 Pa), HFs maintain planar shapes and re-orientate very quickly to the horizontal direction (Figures 10 and 11). Thus, the occurrence of irregular step-like HF geometries is dependent on the combined effect of the differential stress and the gelatin elastic modulus. Additionally, guidance of initial notches on HF propagation becomes extremely limited in a much higher differential stress state ($\sigma_d$ = 2000 Pa), and HF orientations in this situation are dictated by the minimum in situ stress direction.

From the above experimental results, it is concluded that the efficiency of DHF in field application depends on both the differential stress and the rock elastic modulus. More specifically, it is the relative magnitude of the differential stress and the rock elastic modulus that determines HF propagation rather than the absolute magnitudes of these two factors. The initial notch is able to ensure the formation of radial HFs (rather than axial HFs), as evidenced by the experimental results of all the test cases. Hence, the application of DHF in field-scale hydraulic fracturing is important to prevent near-borehole tortuosity of HFs (due to their axial initiation) and can contribute to more regular HF geometries and longer HF radii. This is recognized by Catalan et al.'s [37] and Jeffrey et al.'s [47] field observations at Cadia East Mine (a hard rock mine in Australia) and Narrabri Mine (a coal mine in Australia), respectively. Nevertheless, HFs could propagate asymmetrically if hydraulic fracturing is carried out in relatively weak rock masses under high differential stresses (Figures 6 and 9). In this situation, a closer borehole distance is required to ensure that the rock mass is fully hydraulically fractured. In the mining industry, hydraulic fracturing is performed on either a relatively large scale (e.g., orebody pre-conditioning in hard rock mining [48,49] and hard roof pre-conditioning before the initial mining stage in coal mining [50,51] in which the HF radius is about 20 to 30 m) or a relatively small scale (e.g., caveability control of hanging roofs at the face end in coal mining [52] and hard roof cutting near gob-side entries [53] in which the HF radius is about 3 to 5 m). In the former case, HF propagation is mainly dictated by the minimum in situ stress direction, and borehole arrangement should adjust to the in situ stress condition. DHF in this situation decreases breakdown pressure and favors transverse initiation of HFs [37]. On the contrary, the initial notch in small-scale hydraulic fracturing is necessary for creating orientation-controllable HFs in order to cut the beam-like hard roof at the face end or near the gob-side entry. The HF trajectory can be directionally controlled by the initial notch under a low differential stress condition (relative to the rock elastic modulus), as supported by the experimental results in Test Cases 1-1 to 1-3 (Figures 3–5) and Test Cases 2-2 and 2-3 (Figures 7 and 8).

Note that hard roofs in coal mining that have low caveability and need to be hydraulically fractured are normally strong and massive. This favors the directional propagation of HFs along the notch direction. Therefore, directional roof cutting by DHF is possible if the minimum principal stress direction in the fracturing area (the abutment stress must be considered in coal mining) is horizontal and perpendicular to the entry axis or the ratio of the differential stress to the rock elastic modulus is sufficiently low. Otherwise, the hydraulic fracturing strategy needs to be adjusted to improve the caveability of the roof mass in order for its immediate caving with the advance of the working face (e.g., fracturing an area 20 to 30 m around the mining entry) instead of creating a continuous cutting line to

eliminate the beam-like structure. Recently, DHF extended its application to assist mechanical excavation in strong, massive rock masses (by road-headers) at Guojiawan Coal Mine, China. Normally, pre-conditioning is performed 10 m ahead of the excavation working face, which has a cross-section of about 6.4 m (in width) × 4.2 m (in height). Effective notch cutting favors formation of a set of orientation-controllable radial HFs along the horizontal hydraulic fracturing borehole in order to reduce the block size of the rock mass and hence facilitate hard rock excavation [54].

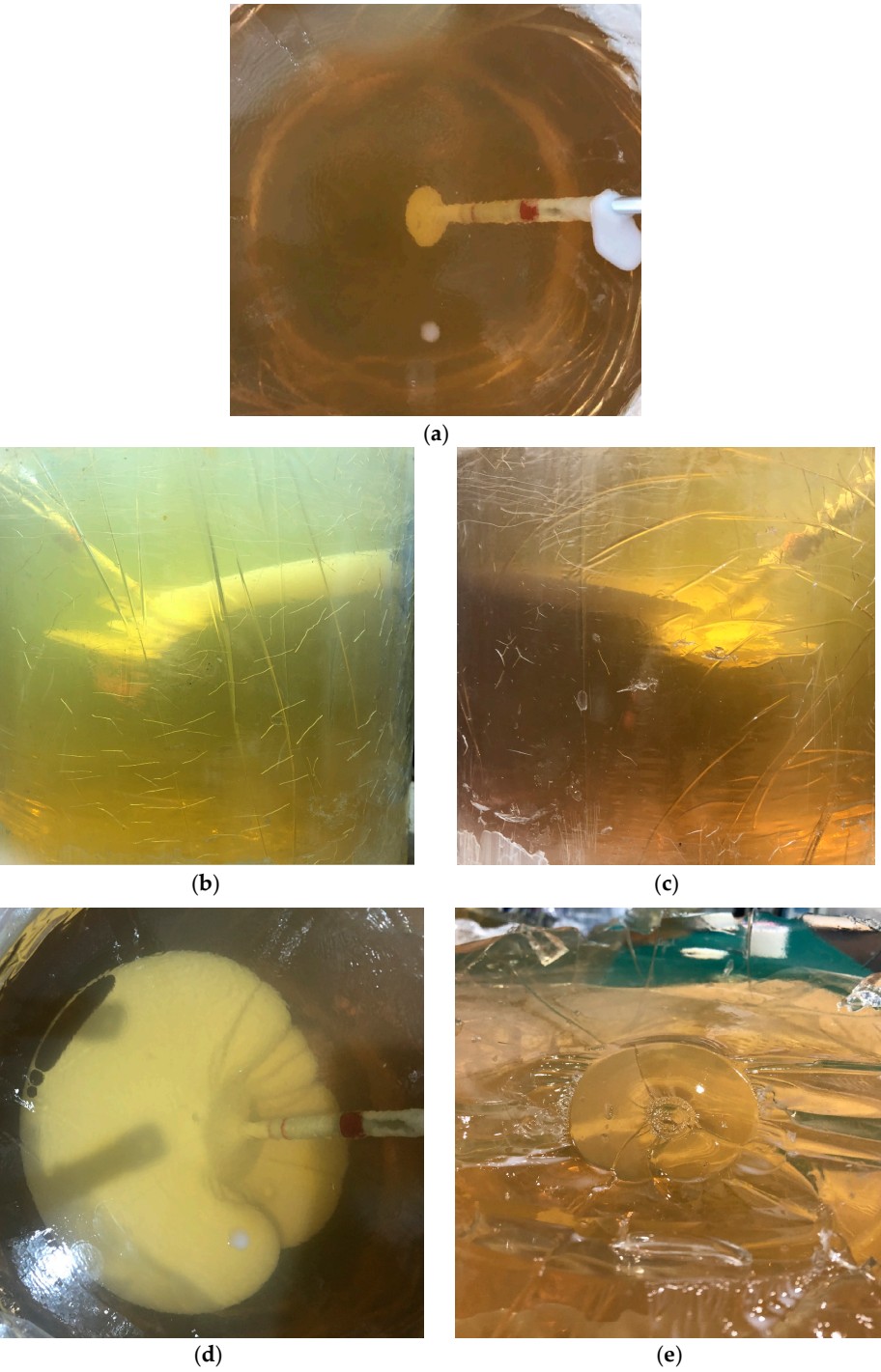

**Figure 9.** Results of Test Case 3-1 ($w = 6\%$ and $\sigma_d = 2000$ Pa): (**a**) HF initiates from initial notch (plan view); (**b**) irregular and asymmetrical HF trajectory; (**c**) irregular and asymmetrical HF trajectory observed from another angle; (**d**) step-like fracture surface; (**e**) smooth notch surface and uneven fracture surface (due to HF re-orientation).

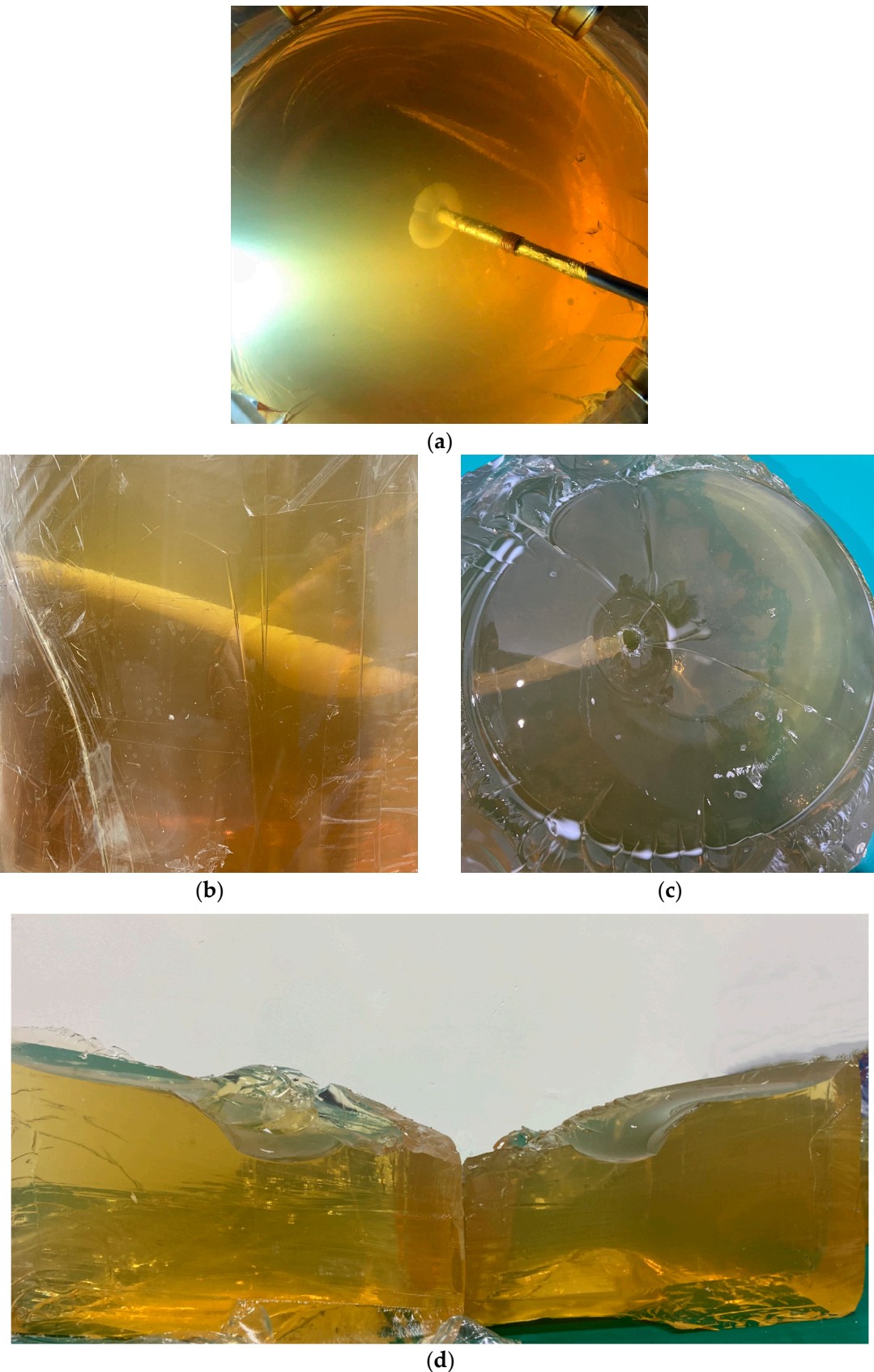

**Figure 10.** Results of Test Case 3-2 ($w$ = 8% and $\sigma_d$ = 2000 Pa): (**a**) HF initiates from initial notch (plan view); (**b**) HF trajectory (section view); (**c**) fracture surface (plan view); (**d**) fractured gelatin sample taken out from container.

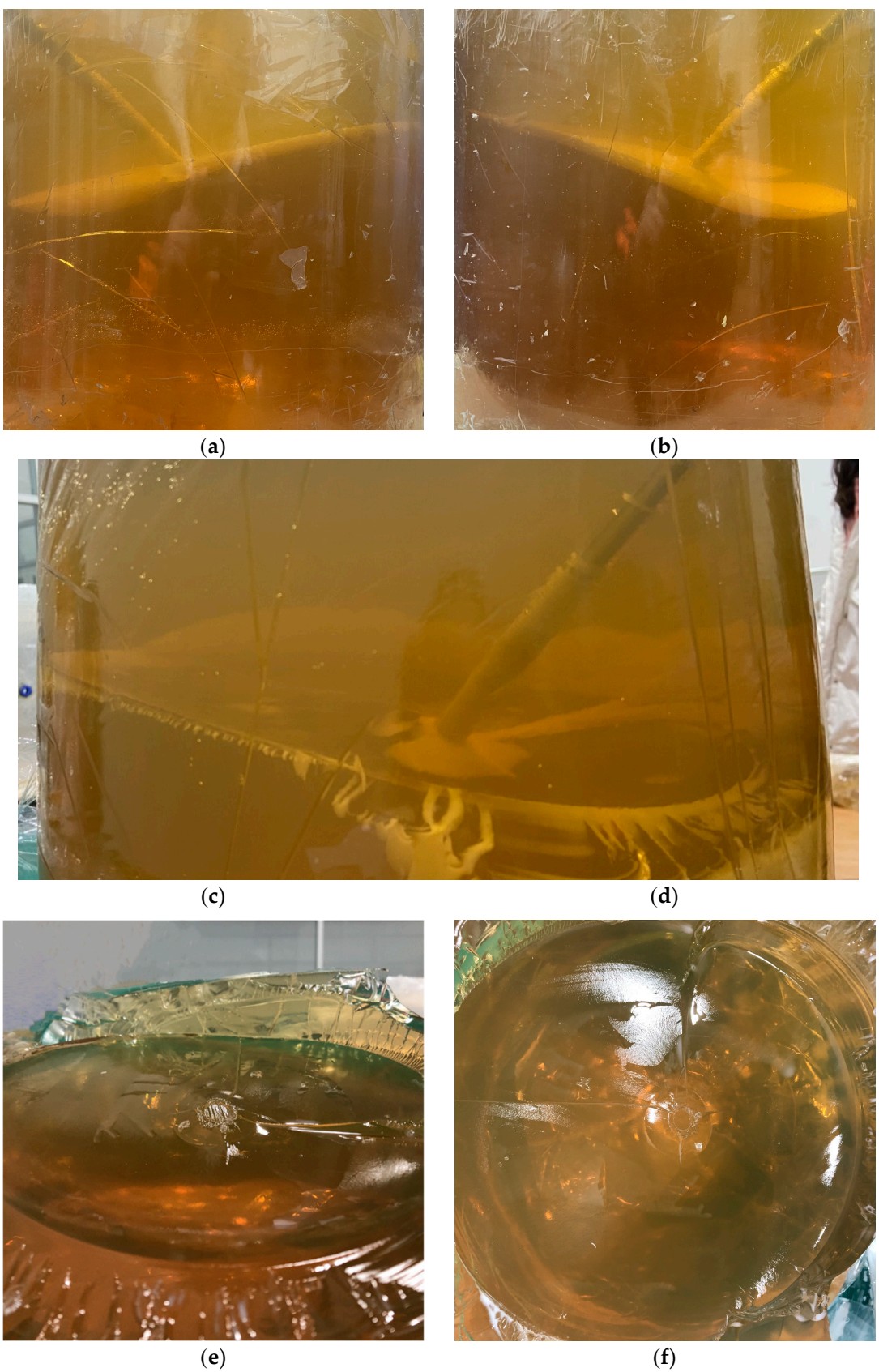

**Figure 11.** Results of Test Case 3-3 ($w$ = 10% and $\sigma_d$ = 2000 Pa): (**a**) HF trajectory; (**b**) HF trajectory observed from another angle; (**c**) fractured gelatin sample taken out from container; (**d**) fracture surface (section view); (**e**) fracture surface (plan view); (**f**) fracture surface (top view).

In summary, the effects of the differential stress and the material elastic modulus on DHF are qualitatively examined and discussed in this study. The correlation between the differential stress and the material elastic modulus will be further quantitatively investigated in future studies in order for successful application of DHF with various purposes in the mining industry.

## 4. Conclusions

In this study, factors affecting HF re-orientation in DHF are analyzed by laboratory experiments in transparent gelatin samples with a self-developed experimental device. The following conclusions are made based on the experimental results:

- It is the relative magnitude of the differential stress and the gelatin elastic modulus that determines HF re-orientation rather than the absolute magnitudes of these two factors. HF re-orientation occurs even if the differential stress ranges from 1000 to 2000 Pa, as long as the gelatin sample also has relatively low elastic moduli.
- Both the re-orientation distance and the geometry of the HF are dependent on the combined effect of the differential stress and the gelatin elastic modulus.
- The HF tends to propagate asymmetrically in gelatin samples with low elastic moduli and under high differential stresses. Asymmetrical, step-like HF geometry is found in the test case that has the lowest gelatin elastic modulus and the highest differential stress, which indicates that both shear failure and tensile failure happen during HF re-orientation.
- HF re-orientation is not necessarily a near-borehole effect, and HFs can propagate along the notch direction for longer distances in stiffer gelatin samples under relatively low differential stresses (1000 Pa and 1500 Pa). For the gelatin concentrations considered in this study (6%, 8%, and 10%), the promotion of DHF by high gelatin elastic moduli is limited once the differential stress reaches 2000 Pa.
- In field applications, DHF is recommended to be conducted on a relatively small scale and in strong, massive rock masses under low or moderate differential stress states. Otherwise, the hydraulic fracturing strategy needs to adjust to the minimum in situ stress direction.

**Author Contributions:** Methodology, B.L.; Writing—original draft, H.Z.; Writing—review & editing, Q.H. All authors have read and agreed to the published version of the manuscript.

**Funding:** The work of this paper is financially supported by the National Key R&D Program of China (2022YFC2905600) and the National Natural Science Foundation of China (51974293).

**Data Availability Statement:** The raw data supporting the conclusions of this article will be made available by the authors on request.

**Acknowledgments:** The corresponding author of this paper would like to thank his deceased grandmother, Chongfen Ran, for the love and care received from childhood.

**Conflicts of Interest:** The authors declared that they have no conflicts of interest to this work.

## Abbreviations

| | |
|---|---|
| DHF | directional hydraulic fracturing |
| HF | hydraulic fracture |
| POF | polyolefin |
| PVC | polyvinyl chloride |

**List of Symbols**

| | |
|---|---|
| $E$ | elastic modulus, Pa |
| $K_c$ | fracture toughness, $\mathrm{Pa} \cdot \sqrt{\mathrm{m}}$ |
| $R^2$ | coefficient of determination, 1 |
| $W$ | gelatin concentration, 1 |
| $\sigma_d$ | differential stress, Pa |

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
