# Peer review of "Visualized Hydraulic Fracture Re-Orientation in Directional Hydraulic Fracturing by Laboratory Experiments in Gelatin Samples"

_applsci, doi:10.3390/app14052047_

Round 1
Reviewer 1 Report
Comments and Suggestions for Authors
General comment: The structure of the article is unclear and inconsistent with the guidelines adopted by the journal. The article should include basic sections such as: 1. Introduction, 2. Materials and Methods, 3. Results, 4. Discussion. Subsection 2.1 could be a separate chapter titled, for example, "Related Works," "Background," etc., or it could be part of Chapter 1. There is a lack of basic information about gelatin and its preparation method. It would be necessary to provide a more detailed explanation of why gelatin was chosen as the material for the study.
Line 6, 7 – add e-mail addresses
Line 51 - The obtained samples are not entirely transparent. To achieve better results, the authors should use gelatin with a higher degree of purity. Please explain why such a choice of research material was made.
General note - Some of the citations are written in superscript. This should be be changed.
Author Response
First of all, the authors should mention that they highly appreciate the editor and reviewers’ comments on the manuscript (MS). With great care, the authors reviewed the comments and suggestions and have incorporated major revisions into the manuscript. Listed below are the authors’ point-by-point responses to the reviewers’ comments. The revisions are highlighted yellow in the revised MS.
1 Comments from Reviewer 1
1.1 Comment 1
The structure of the article is unclear and inconsistent with the guidelines adopted by the journal.
Authors’ Response
We appreciate the reviewer’s suggestion to improve the quality of the MS. A new section named "Related Works " is added.
1.2 Comment 2
There is a lack of basic information about gelatin and its preparation method.
Authors’ Response
We really appreciate the valuable suggestion from the reviewer. The preparation method of gelatin is as follows: Weighing the specific weight concentrations of gelatin powder (i.e. 6 %, 8 % and 10 %) into about 80℃ hot water to fully melt into gelatin fluid. See Line 114 in the revised MS.
1.3 Comment 3
It would be necessary to provide a more detailed explanation of why gelatin was chosen as the material for the study.
Authors’ Response
We appreciate the reviewer’ suggestion and kindly remind that this MS already contains detailed explanations of choosing gelatin samples. The reasons are as follows:
- Most laboratory experiments on hydraulic fracturing are conducted in untransparent materials (e.g. rocks), and the existing experimental results of DHF provided by Deng et al. (2016) only presented the final HF path from a two-dimensional section plane (see Figure 13 in Deng et al. (2016). The complete HF development process (i.e. HF initiation and propagation) and the tortuous three-dimensional HF geometry remain invisible.
- Besides, the simulation of the initial notch (which is the key part of DHF) in Deng et al. (2016) is questionable.
- There are already some researchers using gelatin instead of traditional rock for hydraulic fracturing experiments. Hubbert and Willis (1972) also hydraulically fractured a cylindrical gelatin sample sus-taining radial confining pressure to examine the HF propagation plane relative to the minimum in-situ stress direction. Hence the manual injection of the plaster-of-paris slurry in each test is kept extremely slow to ensure HF propagation is under a quasi-static state, which is consistent with Takada’s (1990) hydraulic fracturing experiment.
1.4 Comment 4
The obtained samples are not entirely transparent. To achieve better results, the authors should use gelatin with a higher degree of purity.
Authors’ Response
We do agree that the obtained samples are not entirely transparent. First of all, the higher the concentration of gelatin, the less transparent it becomes. Second, Plaster-of-paris slurry is taken as fracturing fluid since this material is allowed to set after hydraulic fracturing, thus providing permanent record of HF geometries (Hubbert and Willis ,1972).
Reviewer 2 Report
Comments and Suggestions for Authors
In this paper, the authors have experimentally examined factors affecting hydraulic fracturing reorientation in directional hydraulic fracturing in transparent gelatin samples with a self-developed experimental device. The authors have determined that it is the ratio between the differential stress and gelatin elastic modulus that determines HF re-orientation rather than the absolute magnitudes of these two factors.
The manuscript is generally well-written. Some remarks on improving it are though:
1/ The authors should explain the sample preparation better. No details are provided on the exact materials used (e.g. which type of gelatin, which manufacturer, how they dissolved it, etc.).
2/ The indentation testing device should also be specified (manufacturer, etc.). In addition, the authors should also explain which way they measured the elastic modulus in their indentation tests.
3/ The references should be given in a standard format within the manuscript.
Author Response
First of all, the authors should mention that they highly appreciate the editor and reviewers’ comments on the manuscript (MS). With great care, the authors reviewed the comments and suggestions and have incorporated major revisions into the manuscript. Listed below are the authors’ point-by-point responses to the reviewers’ comments. The revisions are highlighted yellow in the revised MS.
2 Comments from Reviewer 2
2.1 Comment 1
The authors should explain the sample preparation better. No details are provided on the exact materials used (e.g. which type of gelatin, which manufacturer, how they dissolved it, etc.).
Authors’ Response
We really appreciate the valuable suggestion from the reviewer. The exact materials can be seen lines 117 to 119 in the revised MS.
2.2 Comment 2
The indentation testing device should also be specified (manufacturer, etc.). In addition, the authors should also explain which way they measured the elastic modulus in their indentation tests.
Authors’ Response
The indentation testing is conducted to obtain the Elastic Modulus of gelatin, based on the Hertzian contact theory. The test method is consistent with that of Kavanagh et al. (2013) and Ham and Kwon (2019).
2.3 Comment 3
The references should be given in a standard format within the manuscript.
Authors’ Response
We appreciate the reviewer’s recommendation of this MS. Standard formats of citations are revised in this MS.
Reviewer 3 Report
Comments and Suggestions for Authors
Authors must attend to the following points before being considered for publication.
1. In the introduction section, the authors should relate how the physicochemical properties of gelatin relate to the studies and expected DHF behavior.
2. The experimental section must be reorganized, they must include an experimental diagram that allows the reader to have a route of their experimental set, it is very confusing to read.
3. The results must be discussed in a way that sequences one section with the other. There is no connection in the current form.
It is recommended to see the following reference
https://doi.org/10.1016/j.ijrmms.2018.12.023
4. The authors should explain how they would expect this study to be applied to other pure gels or composite polymers.
Author Response
First of all, the authors should mention that they highly appreciate the editor and reviewers’ comments on the manuscript (MS). With great care, the authors reviewed the comments and suggestions and have incorporated major revisions into the manuscript. Listed below are the authors’ point-by-point responses to the reviewers’ comments. The revisions are highlighted yellow in the revised MS.
3 Comments from Reviewer 3
3.1 Comment 1
In the introduction section, the authors should relate how the physicochemical properties of gelatin relate to the studies and expected behavior.
Authors’ Response
We appreciate the reviewer’s recommendation of this MS. The discussion of the physicochemical properties of gelatin and expected DHF behavior is introduced in revised MS (Subsection 2.1. Related Works). Gelatin is homogeneous, isotropic, elastic and brittle. It is recognized as an ideal analogue to rock-like materials in laboratory experiments. Ham and Kwon’s measurement results proved that the initiation pressure and fracture propagation velocity increased with the gelatin concentration. Moreover, the opening width of hydraulic fractures in gelatin is mainly determined by the medium stiffness and fluid pressure (see Lines 97 to 100 in the revised MS).
3.2 Comment 2
The experimental section must be reorganized, they must include an experimental diagram that allows the reader to have a route of their experimental set, it is very confusing to read.
Authors’ Response
We really appreciate the valuable suggestion from the reviewer. We have rewritten the experimental chapter in the revised MS. The experimental device mainly consists of three parts: a container for placing gelatin samples, a pressurizing device and a fracturing device (see Lines 106 to 120 in the revised MS).
3.3 Comment 3
The results must be discussed in a way that sequences one section with the other. There is no connection in the current form.
Authors’ Response
We appreciate the reviewer’s recommendation. We have added some explanations in the revised MS. Actually, the results are discussed chapter by chapter. Secondly, because the statement in our original manuscript is not very clear, we have slightly modified the statement to make readers more able to understand our statement, so as to avoid the misunderstanding that we do not sequence one section with the other.
3.4 Comment 4
The authors should explain how they would expect this study to be applied to other pure gels or composite polymers.
Authors’ Response
We really appreciate the valuable suggestion from the reviewer. Many scholars have used gelatin materials to study the hydraulic fracturing behaviors including the geometry and width of fractures, interaction between HFs and NFs (natural fractures) and so on. We look forward to further research on the stress shadow effect of hydraulic fracturing using gelatin materials or composite polymers.
Round 2
Reviewer 1 Report
Comments and Suggestions for Authors
The authors addressed my comments and suggestions and made appropriate changes accordingly. I believe the article can be published in its current form.
Reviewer 3 Report
Comments and Suggestions for Authors
The authors responded to the reviewers' comments, the manuscript was improved and could be considered for publication.